# Reconciling patterns of long-term topographic growth with coseismic uplift by synchronous duplex thrusting

Yuqing Zhang [1,2,3], Hanlin Chen [2,3] ✉, Xuhua Shi [2,3,4] ✉, Rafael Almeida[5], Richard Walker[6], Xiubin Lin [2,3], Xiaogan Cheng [2,3], Hongdan Deng [2,3], Zhuxin Chen [1] & Xiu Hu [7]

How long-term changes in surface topography relate to coseismic uplift is key to understanding the creation of high elevations along active mountain fronts, and remains hotly debated. Here we investigate this link by modeling the development of growth strata and the folding of river terraces above the Pishan duplex system in the southern Tarim Basin. We show that synchronous duplex thrusting of two neighboring faults with varying slip rates, associated with in-sequence propagation of the Pishan thrust system, is required to explain the presence of opposite-dipping panels of growth strata on the duplex front, and basinward migration of terrace fold crests. Importantly, this process of synchronous thrusting within the duplex reconciles the discrepancy between the deformation of terrace folds at the $10^{-1}$–$10^{0}$ million-year timescale and the maximum coseismic uplift of the 2015 $M_w$ 6.4 Pishan earthquake on the frontal thrust. These results suggest that topography mismatch at different time scales can reflect the long-term kinematic evolution of fault systems. Thus, our study highlights the importance of characterizing complex subsurface fault kinematics for studying topographic growth, and motivates rethinking of the mountain building process in worldwide active fold-and-thrust belts, from short-term to long-term timescales.

Topographic growth reshapes the Earth's surface and affects a wide range of processes such as regional climate patterns and terrestrial biodiversity[1–4]. In the long term, the growth of topography can be driven by crustal thickening either through viscous deformation at depth or brittle deformation in the upper crust over millions of years and is usually described as a continuous, quasi-steady process[5]. In the short term, topography can be produced via repeated episodes of coseismic folding and uplift, in particular along fold-and-thrust belts,

that in aggregate form high-topography mountains[6,7]. The relationship between short-term earthquake uplift and long-term mountain building has been hotly discussed for decades, with many studies arguing that high topography is created through repeated coseismic slip[8–11]; while others look for alternative mechanisms, for instance, from inelastic deformation occurring during interseismic periods, to explain the discrepancy between the positions of repeated coseismic uplift and long-term topographic growth that is sometimes observed (e.g.,

[1]Research Institute of Petroleum Exploration and Development, PetroChina, Beijing, China. [2]Key Laboratory of Geoscience Big Data and Deep Resource of Zhejiang Province, School of Earth Sciences, Zhejiang University, Hangzhou, China. [3]Research Center for Structures in Oil and Gas Bearing Basins, Ministry of Education, Hangzhou, China. [4]Xinjiang Pamir Intracontinental Subduction National Observation and Research Station, Beijing, China. [5]Department of Geological Sciences, San Diego State University, San Diego, USA. [6]Department of Earth Sciences, University of Oxford, Oxford, UK. [7]Guangdong Provincial Key Laboratory of Geodynamics and Geohazards, School of Earth Sciences and Engineering, Sun Yat-Sen University, Zhuhai, China. ✉e-mail: hlchen@zju.edu.cn; shixuhua@zju.edu.cn

the 2005 $M_w$ 6.0 Qeshm Island in Zagros, the 2015 $M_w$ 7.8 Gorkha in Himalaya, and the 2015 $M_w$ 6.4 Pishan in southern Tarim)[12–14].

Previous efforts in examining the relationship between long-term topographic growth and coseismic uplift in fold-and-thrust belts relied mainly on surface records of activity of single faults[12,15,16]. Thrust fault kinematics and geometry (especially for blind thrusts) in the upper crust are in many cases constrained by their relationship with overlying folds[17–20]. However, the subsurface geometry and detailed kinematics of multi-thrust fault systems, such as duplexes, have been less well explored. In several fold-and-thrust belts where these fault systems have been described, synchronous slip on two or more neighboring faults has been documented[21–25]. How to detect whether synchronous thrusting occurs, and how such a process affects surface deformation remains unclear, hampering our understanding of the relationship between medium-to-long-term topographic growth and coseismic uplift.

Located at the northwestern margin of the Tibetan Plateau, the Pishan fold-and-thrust belt is a classic duplex system that accommodates the deformation propagated northward from the Tibetan Plateau into the Tarim Basin (Fig. 1)[26,27]. The tectonic activity of this duplex system has produced a series of earthquakes[28–31], including the 2015 $M_w$ 6.4 Pishan earthquake that occurred at the frontal ramp of the duplex and produced coseismic surface folding (Fig. 1d)[27,32]. However, during this event, the crest of coseismic uplift was misaligned with older fluvial terrace folds above the frontal fault ramp[12]. Since the long-term growth of fluvial terrace folds cannot be fully explained by repeated coseismic slip similar to that observed in 2015[12], the discrepancy between the long-term and present uplift has been partly attributed to other mechanisms such as postseismic slip or inter-seismic inelastic deformation, similar to other active fold-and-thrust belts[13,14,33,34].

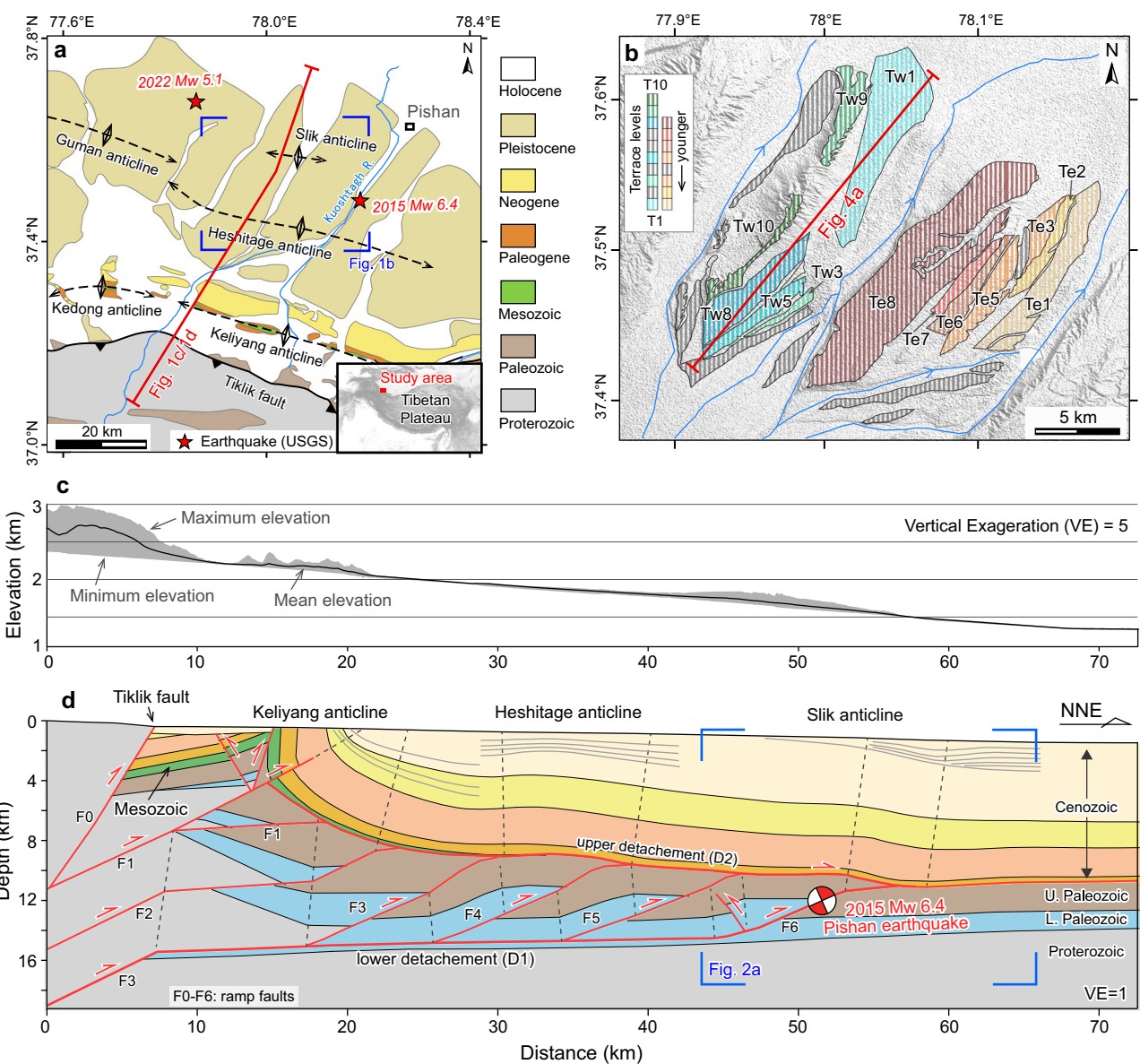

**Fig. 1 | Geological context, surface topography and geologic cross-section of the study region. a** Geologic map of Pishan belt; modified from geological maps in the Xinjiang Uygur Autonomous Region with a scale of 1:500,000. **b** Geomorphologic mapping of fluvial terraces in the Pishan belt. The terrace levels were adapted from Ainscoe et al.[12]. Elevation data are from the 12.5 m-resolution digital elevation models. **c** The topographic profile along a swath of 1 km in width and **d** Geologic cross-section across the Pishan fold-and-thrust belt (location in (**a**) and see Fig. S6 in Supplementary Information for the seismic reflection profile). The location and depth of the Pishan earthquake from He et al.[58]. Note: the Cenozoic strata shown in the geologic map of (**a**) are not correlated with those in the cross-section in (**d**).

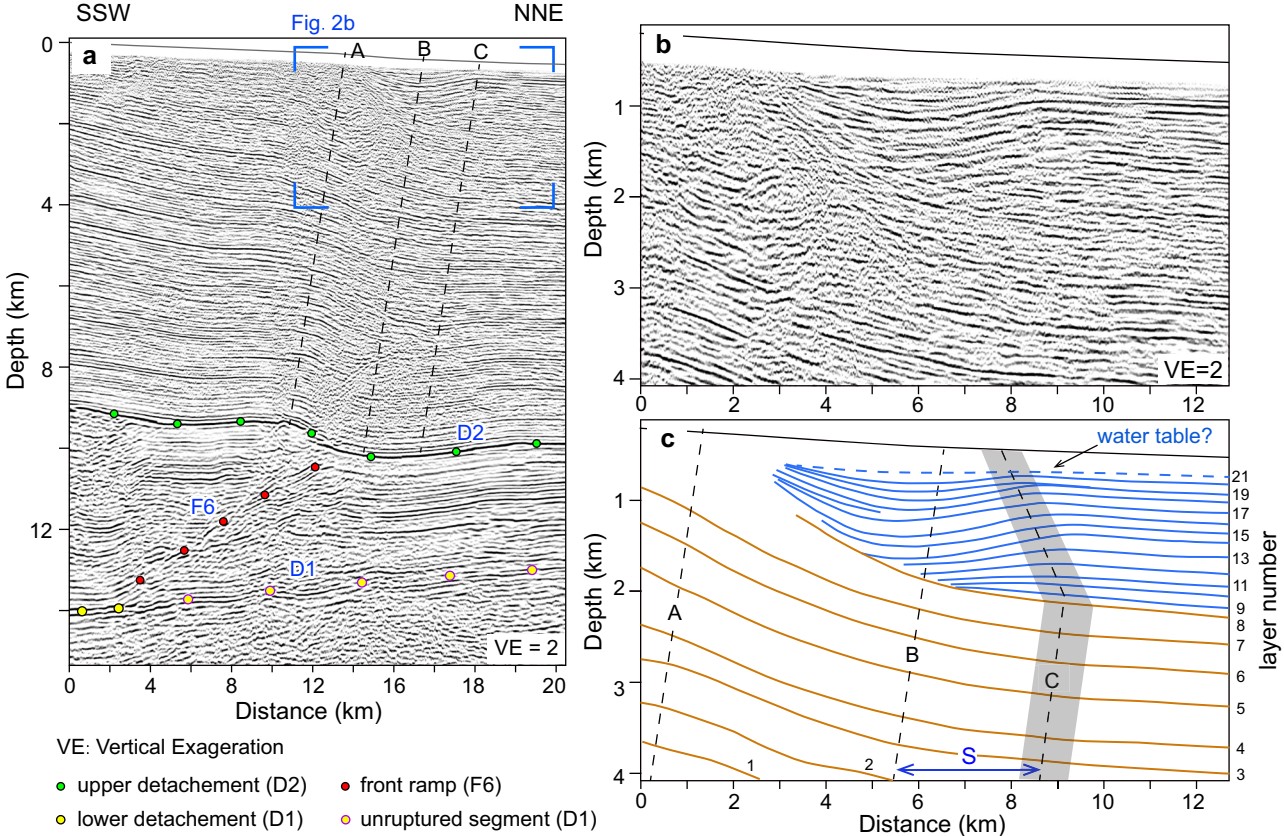

**Fig. 2 | The seismic reflection profile and interpretation of the Slik Monocline.**
**a** Uninterpreted seismic reflection profile. **b** Enlarged area of the seismic profile shown in the blue box in (**a**). **c** Line drawing of (**b**). Yellow lines represent the pre- growth strata; blue lines represent the growth strata. S denotes the amount of fault slip across the roof fault since the frontal ramp initiated.

In this study, we re-interpret a high-resolution seismic reflection profile to describe in detail the geometry of the Pishan duplex system, as well as of a newly identified interval of overlying growth strata. We then combine our observations of the growth strata geometry with previously published terrace folding measurements to develop new kinematic models to describe their formation with the goal of characterizing the evolution of the Pishan duplex. We show that synchronous thrusting of two adjacent faults in the duplex system is required to form these structures. Finally, we propose an alternative explanation for the location mismatch between coseismic uplift and terrace folds, demonstrating how complex subsurface processes play a critical role in generating topography in active fold-and-thrust belts at various time scales.

## Results

### Observations from the Pishan seismic reflection profile

The geometry of the Pishan fold-and-thrust belt is well imaged by an NNE-striking seismic profile (Fig. 2a, see Supplementary Text S2 for details). This profile can be separated into three sequences by two strong continuous seismic reflections (D1 and D2)[26]. From bottom to top, the sequences are (1) below D1: a discontinuous and chaotic reflector package denoting the pre-Cambrian basement[26,35], (2) between D1 and D2: a continuous and southward thickening reflector package representing Paleozoic strata, and (3) above D2: a continuous and generally north-dipping reflector package representing Cenozoic strata, which comprises the Slik monocline at the duplex front. D1 and D2 are also the lower detachment (rooted in the lower Cambrian), and upper detachment (rooted in the basal Paleogene evaporate), of the Pishan duplex, respectively. The strata between them is deformed by four ramp faults (Fig. 1d) that

comprise a duplex thrust system where in-sequence thrusting has been interpreted[36].

Noticeably, the shallow strata in the forelimb of the Slik monocline feature south-dipping reflections above the overall north-dipping reflections that we interpret as an interval of syn-tectonic strata. The lower part of the south-dipping reflections rests on chaotic reflections (L9-14 in Fig. 2c). Moving upward through the stratigraphy, these south-dipping reflections gradually change into a concave-up synform (L14-20 in Fig. 2c). In the shallowest part, the synform reflections become nearly horizontal (L20-21 in Fig. 3c). Generally, these south-dipping and concave-up reflections are spatially restricted by axial surface A to the south and axial surface C to the north, and do not extend to the southern limb of the Slik monocline. The strata in the synform are also slightly thicker than the undeformed strata to the north (Fig. 2). These growth strata record underlying faulting events, but are not produced by fault-bend folding (Figs. S3a and S4a) or by folding above a tectonic wedge (Figs. S3b and S4b), two basic kinematic processes that are generally considered to dominate at thrust fronts[19,37].

### Modeling results

We use MOVE software to construct novel models of growth strata and terrace fold formation to decipher the subsurface process responsible for the geometries observed above the Slik Monocline, as well as its effect on surface topographic expression. The detailed modeling processes and parameters are provided in the Methods and the Supplementary Information sections.

Growth strata are modeled at the front of a duplex system characterized by synchronous motion of the root and roof thrust faults as they reach the front of the duplex. We use fault models with smooth

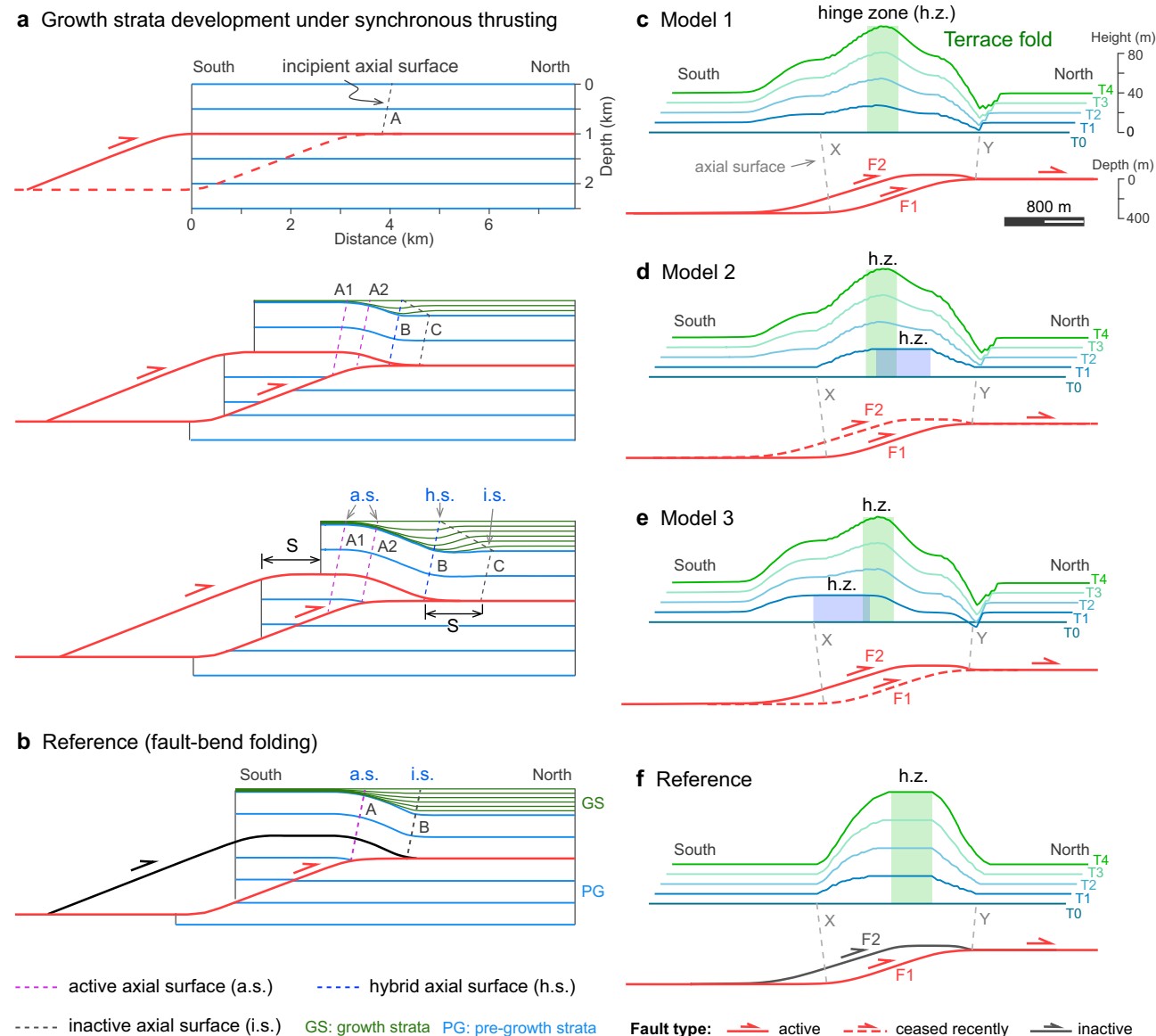

**Fig. 3 | Modeling of growth strata and terrace folds. a** The syn-sedimentary model of synchronous thrusting in the duplex system front. S denotes the amount of basinward fault slip across the roof fault. Note: the hybrid axial surfaces represent inactive axial surfaces (thin black dash lines) for ramp thrusting and active axial surfaces (thin red dash lines) for roof thrusting. **b** The reference model of fault-bend folding[36]. **c–e** The models of terrace folding controlled by two ramp faults in a duplex thrust system. Blue-green lines (T0-T4) represent terraces developed at different periods, with T0 being the youngest, dashed red faults become inactive, solid red faults remain active throughout. See the text for details. **f** The reference model of terrace folds controlled by one ramp fault. Red faults are active, black faults are inactive. The terrace folds in above models (**c–f**) have been shown with vertical exaggeration (VE) of 10.

fault bends, similar to natural examples. The resulting growth strata can be separated into three segments by axial surfaces B and C (Fig. 3a). From north to south, these are (1) an undeformed zone limited by axial surface C to the south, (2) a south-dipping zone located between axial surfaces B and C, and (3) a horizontal to gently north-dipping zone limited by axial surface B to the north. The south-dipping zone rests unconformably on the horizontal pre-growth strata and tapers upward, while the zone with horizontal to gently north-dipping layers onlaps the forelimb of the underlying anticline. These oppositely dipping zones comprise a broad synform at the front of the duplex system. In addition, the growth strata of the synform are systematically thicker than those in the undeformed zone. A similar geometry also forms in models with sharp fault bends and hinge zones (Supplementary Fig. S2).

Such a geometry significantly differs from growth strata in thrust fronts formed by fault-bend folding or wedge structures

(Supplementary Figs. S3a, b and S4a, b)[37,38], but is similar to those related to listric normal faults (Supplementary Figs. S3c and S4c)[38,39]. This growth strata geometry reflects the contribution of the roof thrust during the synchronous thrusting of a duplex system. Here, the downward slip of roof thrust between axial surfaces A and B results in the thickening of the growth strata that is initially onlapping on the north-dipping forelimb. Meanwhile, these thickened strata gradually turn to south-dipping as the roof thrust carries them through axial surface B onto the horizontal fault segment. This is similar to the migrating growth strata that is produced as basal detachment fault slips over a bulge on an otherwise planar fault segment[40]. Therefore, the horizontal distance between axial surfaces B and C represents the amount of fault slip along the roof fault since the frontal ramp initiated slip (Fig. 3a and Supplementary Fig. S2).

We model terrace folds in a duplex system characterized by synchronous motion of two ramp thrusts with three scenarios: (1) two

ramps active with constant slip rates (Model 1; Figs. 3c), (2) the rear ramp slip ceasing in the latest stage (Model 2; Fig. 3d), (3) the frontal ramp slip ceasing in the latest stage (Model 3; Fig. 3e). We compare models of synchronous thrusting to a reference model with a single ramp. Our results show that synchronous thrusting models have wider terrace-fold wavelengths and a slightly more hinterland-ward distribution of fold crests (Fig. 3c–e) compared with the reference model, due to the contribution of the rear ramp (F2) (Fig. 3f). Moreover, variations of slip rates among the two ramps leads to the migration of crests of lower (younger) terrace folds relative to the higher (older) ones. Model 2 shows that crest axes of lower terrace folds migrated forward (northward) relative to those of the higher terrace folds as the rear ramp (F2) slip ceases (Fig. 3d); while Model 3 shows the opposite migration of fold-crest axes as the frontal ramp (F1) slip ceases (Fig. 3e). In contrast, Model 1 with temporally constant slip rates for the two ramps generates nearly the same fold-crest axes (Fig. 1c).

Modeling results also show a small trough on the forelimb of the terrace folds (axial surface Y in Fig. 3c–e) during the synchronous duplex thrusting. These concave-up terraces, analogous to the syn-form in the growth strata (Fig. 3a), are products of downward slipping of the roof thrust at the forelimb of the anticline, which could lead to the intersection of higher terraces with lower ones in a cross-section view (Fig. 3e). However, this concave-up topography is probably compensated by sediments in the field.

### Comparison of the observations with modeling results

By comparing our observations across the Pishan duplex belt with our modeling results, we find that these south-dipping and thickened strata in the forelimb of the Slik monocline are consistent with the geometry of our growth strata model (Figs. 2 and 3a). This suggests that even though the frontal ramp is the most active fault in the Pishan belt, building the Slik monocline and generating a series of recent earthquakes[12,41], the roof thrust must have remained active, at least for a period of time, since the frontal ramp was activated. Therefore, the frontal ramp and roof thrust were active synchronously in the long term, although whether the slip is strictly synchronous or movement alternated between the two faults remains unknown. Nevertheless, we propose that most of, if not all, the slip on roof thrust comes from the penultimate ramp fault (F5 in Fig. 1d). This proposition is supported by two observations: (1) the sequential younging of growth strata related to anticline formation towards the foreland across the Pishan belt, as documented in this study (Fig. 1d and Supplementary Fig. S7) and by Liang et al.[36], indicating in-sequence thrusting in this duplex thrust system with recent deformation concentrated on the front two ramps, and (2) the presence of uplifted terraces above the first two ramps instead of the antepenultimate ramp (F4), where the crest of an older fold (the Heshitage anticline) is located (Figs. 1 and 4).

The fluvial terrace deformation across the Pishan belt has been well documented after the 2015 $M_w$ 6.4 earthquake (Fig. 1b)[12,41–43]. Studies show that fold crests of the lowest (youngest) terraces are about 3–4 km north of the fold crests in the higher (older) terraces and that the forelimb of the fold in lower and higher terrace flights (Tw1 and Tw9) all intersect (Figs. 1b and 4a)[12,41]. This terrace geometry corresponds to Model 2 of our terrace modeling (Fig. 3d), which indicates that the penultimate ramp ceased or decreased in slip rate relative to the frontal ramp of the Pishan belt during the latest phase of deformation. This inference of variation in slip rates along the two frontal ramps is supported by the shallowest growth strata in the forelimb of the Slik monocline, which becomes nearly horizontal overlying the concave-up strata and probably documents the cessation of slip on the roof thrust (Fig. 2b, c and Supplementary Fig. S8). This result is also consistent with accelerating incision rate across the belt since ~400 ka, a finding that was interpreted to signify a transition from deformation partitioning between the two ramps to deformation concentrating mainly on the frontal ramp[41].

Moreover, dating results show that the youngest terrace (Tw1) in Fig. 4a was abandoned at 220 ka[12,41], which means the lowest terrace was deformed after the cessation of the penultimate ramp (~400 ka), if the previous dating is correct[41]. Our modeling results of terrace folds above the frontal ramp by fault-bend folding, are generally consistent with the geometry of terrace Tw1 when the fault slips 100 m (Fig. 4a, see Supplementary Text S3 for details). This consistency suggests that the lowest terrace fold was exclusively controlled by the frontal ramp. Collectively, the Pishan belt documents coeval fault slip on the frontal and penultimate ramps for a period of time that transitions to slip only on the frontal ramp. This synchronous thrusting process probably represents the transition of deformation in an in-sequence duplex system where the hanging-wall ramp does not cease immediately at the onset of faulting along the footwall ramps (e.g., refs. 44,45). In other words, both ramps slip coevally until the older one is abandoned. During this process, the lower (younger) terrace folds gradually migrate basinward relative to the main fold crests due to the deactivation of the penultimate ramp (Fig. 4).

## Discussion

The 2015 $M_w$ 6.4 Pishan earthquake ruptured the frontal ramp of the duplex and caused folding in the southern Tarim Basin (Fig. 1)[32]. During this event, however, the peaks of coseismic and postseismic uplift were shifted northward by ~5 km and ~7 km, respectively, with respect to the highest terrace fold crests above the frontal ramp (Fig. 4a)[12]. This discrepancy indicated that the terrace folds could not be fully explained by repeated coseismic and postseismic slip in events such as the 2015 Pishan earthquake, assuming geometric models that generate surface folds with slip on single seismogenic reverse faults beneath them. This led Ainscoe et al.[12] to suggest that additional mechanisms besides coseismic slip are required to produce the long-term topography in the Pishan fold-and-thrust belt. Here, our study indicates that synchronous activity of the frontal and penultimate ramps before 400 ka likely controlled the highest terrace fold growth, and therefore is impossible to align with the coseismic uplift crest that was produced exclusively by slip along the frontal ramp. In other words, the synchronous duplex thrusting recorded in the growth strata reconciles the shift between the long-term topographic growth observed in the folded terraces and the uplift in the Pishan duplex system caused by the 2015 earthquake.

The amount of shift of long-term vs. short-term uplift is expected to become smaller for the younger (lower) terrace folds as the penultimate ramp gradually deactivates (Figs. 3d and 4a). Indeed, the shift between the peak coseismic uplift of the 2015 Pishan earthquake and the crest of the lowest terrace fold, both above the major part of the frontal ramp, is only ~2 km. This shift can be exclusively controlled by activity of the frontal ramp over the timescale (~220 ka[39]) of the development of the lowest terraces (Tw1). The cause for this shift can be small-scale fault heterogeneities or changes in fault dip and axial plane position that may result in differences in rupture widths and positions of rupture terminations along the frontal ramp, which would in turn lead to variations in positions of peak coseismic uplift during individual earthquakes. For example, above the Puente Hills blind thrust (beneath Los Angeles, CA), the crests of three discrete coseismic folding events are not co-located, but migrate for hundreds of meters from earthquake to earthquake on the forelimb of the fault-related fold[10]. Thus, it is not required that axial surfaces of individual coseismic folds be exactly consistent with those of long-term folds, even though the fold grows solely through the aggregate of these earthquake events.

Collectively, the shift between the Pishan earthquake uplift and terrace fold crests reflects the contribution of the penultimate ramp during the synchronous duplex thrusting, irrespective of specific slip mechanisms (e.g., seismic versus aseismic) along the faults. Thus, at least in theory, the topographic growth in Pishan can be produced

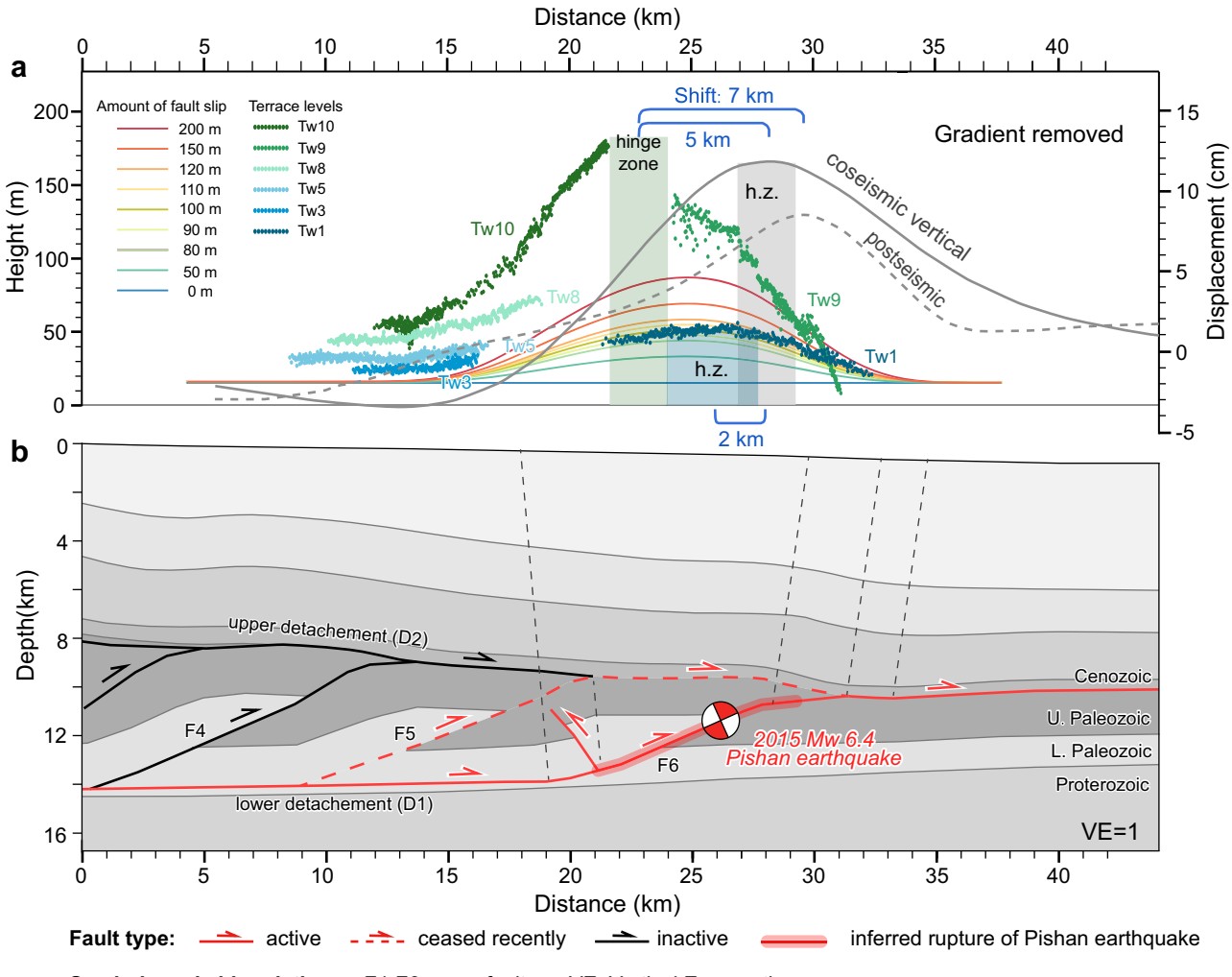

**Fig. 4 | The kinematic relationship in the Pishan fold-and-thrust belt and associated surface deformation. a** Profiles of river terraces, coseismic and post-seismic displacement of the 2015 $M_w$ 6.5 Pishan earthquake from Ainscoe et al.[12]. The shaded vertical bars show the hinge zones of folded terraces. The curved lines represent the modeled terrace deformation above the frontal ramp (details in Supplementary Text S3 and Fig. S9). **b** Geometry of the Pishan fold-and-thrust belt shows the kinematic configuration with the active and inactive faults, and the Pishan earthquake rupture. The rupture data is from Ainscoe et al.[12].

mainly by repeated individual coseismic earthquake motion in a duplex system, without the need to invoke additional mechanisms. Indeed, the postseismic moment accumulated in the 1.5 years following the 2015 Pishan earthquake only accounted for ~4% of the coseismic geodetic moment and is therefore negligible for the topographic uplift[46]. Thus, our study suggests that repeated earthquakes produced by the transition from synchronous thrusting events to slip solely on the frontal thrust of the Pishan duplex can fully explain the misalignment of topography created at the 0.1–1 million-year timescale with the coseismic uplift. This highlights the significance of understanding not only the present subsurface geometry and kinematics of active fold-and-thrust belts but also their evolution for deciphering mechanisms of long-term topographic growth.

Contrary from the classic thrust sequence in the fold-and-thrust belts, where the older fault becomes inactive as the slip is transferred to the new fault[21,44,45], both analog and numerical models show that coeval slip on several fault ramps in a thrust system is common[22,25,47,48]. This synchronous thrusting process has also been reported in worldwide fold-and-thrust belts, including the Himalayas, Pamir, Andes, and North American Cordillera[21,23,24,49–51]. In critical taper models, synchronous thrusting also plays a relevant role, as movement alternating between the fault ramps within the thrust system is predicted to occur to reach the critical angle. This in turn allows the deformation to

propagate forward along the basal decollement[52–55]. Therefore, long-term synchronous thrusting is likely more common than simplified in-sequence models of fold-and-thrust belts' evolution would predict. The novel models of growth strata and terrace fold development presented in this study clearly demonstrate how such a thrusting process may affect surface topographic growth and how detailed subsurface observations can allow us to decipher the evolution of slip transfer on these deep structures. These models can be broadly applied to duplex systems in foreland regions and enable the detailed discussion of the relationships between the thrusting process and surface (and shallow subsurface) responses.

These relationships can further advance our understanding of the mechanism of topographic growth. In the Zagros, eastern Turkey, and Himalayan fronts, the misalignment between topographic folds and earthquake uplift crests has invoked the debate about the relationship between long-term surface topography, subsurface deformation, and coseismic uplift in these regions[6,11,13,14,33,34]. Our novel models suggest that this misalignment may be caused by complex subsurface processes, such as synchronous duplex thrusting, in the active mountain fronts, irrespective of specific slip mechanisms along the faults[13,16,34,56,57]. This highlights the need for detailed knowledge of the spatial and temporal evolution of the thrusting process when using surface deformation records to reveal the mechanism of topographic

growth. Moreover, our results show that when there is synchronous duplex thrusting, classic fault-related folding models with single faults cannot describe the quantitative relationship between terrace folds, growth strata geometry and the underlying thrust faults. Fortunately, our model allows us to describe diagnostic features that can allow us to infer the presence of synchronous duplexing and thus interpret surface deformation features more accurately.

## Methods

### Growth strata modeling

We use the 2D MOVE software to model the development of growth strata and terrace folding above a duplex system where two adjacent ramp faults transfer slip synchronously. Growth strata are constructed on the front of the duplex system, which is characterized by upward motion over the ramp of the root thrust and downward motion over the roof thrust as it reaches the front of the duplex (Fig. 3a). To model the synchronous duplex thrusting, we design a frontal ramp and a roof thrust, both with constant slip rates, for simplicity. The sedimentation rate is approximately equal to the uplift rate, which is similar to the area of our case study in Pishan. Moreover, this kinematic model was produced using fault-related folding models with smooth (Fig. 3a) and sharp (Supplementary Fig. S2) fault-bends[19,37] that assumes no internal deformation of the thrust sheets. In addition, we show growth strata obtained using a model for slip along a single fault in a duplex[37] as a reference (Fig. 3b). The details of growth strata models are presented in the Supplementary information (Text S1.1, Table S1, and Fig. S1).

### Terrace fold modeling

Compared to growth strata, terrace folds preserve deformation records reflecting the subsurface kinematic process over a much shorter time-scale, but are more sensitive. Here we constructed terrace folds in a duplex system with two detachment layers and two ramp faults in between. Three end-member models (Fig. 3c–e) are presented to examine the relationship between terrace folding and synchronous duplex thrusting, using fault-related folding principles[19,37]. The models feature (1) two ramp faults active with constant slip rates (Model 1; Fig. 3c), (2) the rear ramp slip ceasing in the latest stage (terrace T1 was abandoned, Model 2; Fig. 3d), and (3) the frontal ramp slip ceasing in the latest stage (terrace T1 was abandoned, Model 3; Fig. 3e). In Models 2 and 3, as slip on one ramp ceases, the other accelerates in slip rate because the total slip rates are constant. Moreover, we simulate the terrace deformation under single-ramp faulting, as a reference (Fig. 3f). The details of terrace fold models are presented in the Supplementary information (Text S1.2, Table S2, and Fig. S5)

## Data availability

The uninterpreted seismic reflection profile used in this study is provided in Supplementary Fig. S6. The geological data used to compile the geological map of Pishan (Fig. 1a) are obtained from http://www.ngac.org.cn, which is open to the public. The 12.5 m-resolution digital elevation model data used in the geomorphologic map (Fig. 1b) and topographic profile (Fig. 1c) are obtained from https://search.asf.alaska.edu. Modeling of growth strata and terrace fold are based on the MOVE™ software (PE Limited), available at https://www.petex.com. The original modeling results in MOVE format are provided in the Source Data. Source data are provided with this paper.

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

## Acknowledgements

We express our gratitude to Tao Li for fruitful discussions and suggestions. This work was supported by the National Natural Science Foundation of China (Grants No. U22B6002, H.C., 51988101, H.C., 41972217, X.L., and 41972227, X.S.), the PetroChina Major Science and Technology Project (Grants No. 2023ZZ0202 & 2021DJ0301, Z.C.), the Second Tibetan Plateau Scientific Expedition and Research of China (Grant No. 2019QZKK0708, H.C.), and the National Key Research and Development Program (Grant No. 2022YFC3003704, X.S.).

## Author contributions

Y.Z., H.C., and X.S. conceived the study. H.C. directed the project. Y.Z. constructed the growth strata models. Y.Z. and X.H. constructed the terrace fold models. Y.Z., X.S., R.A., and R.W. compared the long-term deformation to the short-term coseismic deformation. Y.Z., X.L., X.C., H.D., and Z.C. interpreted the seismic profile. Y.Z., X.S., and H.C. wrote the original manuscript. All authors discussed, commented, and edited the manuscript.

## Competing interests

The authors declare no competing interests.
