## [Peer Review File · Nature Communications]

Reconciling patterns of long-term topographic growth with coseismic uplift by synchronous duplex thrustingEditorial Note: Parts of this Peer Review File have been redacted as indicated to remove third-party material where no permission to publish could be obtained.

REVIEWER COMMENTS

Reviewer #1 (Remarks to the Author):

Understanding how the high relief along active mountain fronts (e.g., around the Tibetan Plateau and Tian Shan in Asia) achieved is a key question in continental tectonics. Yet, there exists a long-standing debate on the discrepancy between short-term earthquake uplift and long-term topographic growth, in particular, for fold-and-thrust orogenic belts. In this study, Zhang et al. utilized novel models of developing growth strata and terrace folds by synchronous thrusting in a duplex system. Then the authors applied the model results into the Pishan thrust belt where show obvious discrepancy between positions of long-term topographic highs and transient earthquake uplift as recorded by the high-precision InSAR results. The synchronous thrusting explained very well the discrepant positions of long-term topographic highs (folding crests) and transient maximum earthquake uplift, and provides a new and alternative mechanism for the topographic growth in orogenic belts throughout the world, across different timescales. To my knowledge, this is an exciting and novel finding that will be of broad interest to the research community of continental tectonics, geomorphology, geodesy and even the paleo-environment. Moreover, the manuscript text has been well written, and figures well illustrated. Given the importance and high impact of this study, I strongly support its publication in Nature Communications.

Here I have some minor comments on the figures and the reference format, for the authors to consider.

Fig. 1. The interpretation of thrusting structure under the Keliyang anticline seems to be inconsistent with the offset across the strata. Although this minor problem will not affect the authors' interpretation on the synchronous thrusting under the Pishan belt, it needs to be revised.

Please place the horizontal axis labels under the numbers, cite the references for the two earthquakes.

Fig. 2a: "syn-sedimentary model" should be "Growth strata development under synchronous thrusting"; Please show horizontal and vertical axis labels in Fig. 2.

Fig. 3: The labels for horizontal axes of Fig. 3b & 3c should be placed under the numbers and aligned in the middle. Clarify the numbers (3, 4, 5, ..., 15, 17, 19, 21) on the right edge of Fig. 3c in the figure caption, layer numbers, right?

Fig. 4: Reorganize the fault legend at the bottom of the figure as: Fault type: active, ceased recently, inactive.

Also, the format of the references (e.g., Refs. 1 - 3) needs to be double checked and revised according to the journal requirement.

Reviewer #2 (Remarks to the Author):

In their paper entitled "Reconciling long-term topographic and transient seismic uplift by synchronous duplex thrusting" Zhang and co-authors investigate how the topography in the southern Tarim basin grew and how it relates to the Pishan earthquakes (2015, Mw 6.4) and deep duplex structures. The authors mainly conclude that the surface topography (terraces) is due to slip on a deep blind thrust fault of a buried duplex. They also concluded that the recent Quaternary topography was made by the cumulative deformation/uplift of repeated earthquakes on this blind fault.

The question addressed in this study is critical. The deformation model is robust, and model results are convincing. The model nicely explains the growth strata at depth and the geometry of the terraces at surface suggesting that the deformation mechanism proposed by the authors is realistic.

However, the main conclusion of the paper that: "the topographic growth in Pishan can be produced mainly by repeated individual earthquakes in a duplex system, without the need to invoke additional mechanisms, in certain cases." is speculative since not supported by any new data/model etc. It moreover contradicts the previous work by Anisocoe et al. (2017). These authors using InSar, analyses of the same seismic profile together with terraces topography, have already deeply investigated the deformation associated to the Pinshan earthquakes and how it relates to the quaternary topography. Yet, they concluded: "over the timescale of multiple earthquake cycles, interseismic and additional aseismic deformation must be invoked". Yet, to reach this conclusion, Anisocoe et al. modeled the "ground displacements from elastic forward models of earthquakes on the detachment". Anisocoe et al. were "not able to account for the anticline by coseismic slip alone". In contrast, to support their assertion Zhang et al. only speculate (see L212 to 233). Other mechanisms than elastic deformation during repeated earthquakes should be discussed and investigated. This includes interseismic, aseismic deformation, creeping, analestic buckling/folding etc. Those mechanisms are overlooked by Zhang et al. while they could play a major role in explaining the observed uplift of the terraces at surface and the misfit with coseismic displacement observed during the Pishan earthquakes. Consequently, the statement that they "reconcile the discrepancy between the long-term topography of terrace folds and maximum co-seismic uplift of the 2015 Mw 6.4 Pishan earthquake on the thrust front" is a bit overrated (so the title of the paper is).

Moreover, the novelty of this study is questionable which may be an obstacle for a publication in Nature Com. Indeed, most of the data presented in Zhang et al.'s paper were actually presented by Anisocoe et al. (2017) and these authors already reached similar conclusions about the link between long term topographic growth, the Pishan earthquakes and deep deformation mechanisms. They said that: (1) "the coseismic ground displacements from the earthquake do not align with the anticline in Quaternary fluvial terraces", (2) "Our results show that distributed postseismic relaxation of the Cenozoic sediments after the coseismic stress change also cannot explain the anticline growth" and hence (3) "The remaining mechanisms are therefore slip on different parts of the ramp and flat (particularly interseismic transport on the detachment) and internal deformation of the wedge. Although only a few key faults have been interpreted in Figure 6, both the wedge geometry and the reflections show that it has undergone internal thickening in the past and may still be doing so."

However, it is true that Anisocoe et al. remain elusive regarding the deep mechanisms of deformation

within the wedge that are responsible for the Quaternary topographic growth observed at the surface and inferred from terraces topography. Zhang et al explore more deeply these mechanisms using Move software modeling and reached convincing and elegant results. So, the main input of Zhang et al. works is a deformation mechanism at depth.

At last, the organization of the paper is to my opinion awkward. The authors first describe a structural model of duplex and propose different scenarii with different kinematics/timing of fault activation. Syn-tectonics layers are then deposited above the duplex and deformed accordingly to the scenarii. Finally, the authors compare the geometry of these deformed layers to the growth strata observed on the seismic line and terraces geometry measured at the surface. But a logical presentation would have been first to describe the Pishan duplex system and the geometry of the syn-tectonics layers. Then, based on these observations, a structural model the deep duplex structures could be proposed accordingly, and different fault kinematics could be tested. The results of the modeling can be then compared to the reality to discriminate between the scenarii.

In conclusion, I think the results of this paper about the deep long-term mechanisms of deformation are good and very interesting, even if they are, to my opinion, not well presented/organized. But I don't think this study has reached the level quality requested for publication in Nature Communication because it lacks novelty and, more importantly, robust data/model to support the main conclusion that the topography was made only by repeated earthquakes.

I therefore suggest rejecting the paper.

Reviewer #3 (Remarks to the Author):

Review of the manuscript "Reconciling long-term topographic growth and transient seismic uplift by synchronous duplex thrusting" by Zhang and co-authors, submitted to Nature Communications.

In their paper, the authors establish geometries of growth strata related to ramp activity in duplex structures and compare them to seismic lines across the Pishan thrust system in the Tarim Basin. Furthermore, they model and investigate terrace folding to track shorter term topographic evolution.

The manuscript is well written and the topic is of interest for a broad scientific community. They demonstrate how a rather simple technique (modelling of growth strata and terraces with Move for example) can help understanding complex problems such as long-term versus transient deformation in fold-and-thrust belts. Overall, I think the paper is a nice contribution and may be published, however, after some re-organization of the text. At times, I had difficulties to follow the argumentation and in my view the study would gain a lot if, for example, the study area (incl. Fig. 1) was introduced earlier.

I hope my comments are useful and constructive

Comments:

1) The study deciphers differences between long-term and transient deformation. As far as I understood, the authors interpret deformation along duplex thrusts as transient and bulk pure shear deformation of the lithosphere as long-term. Maybe I understood that wrongly, but then it should be explained more clear. Slip along the ramps can also be aseismic, so I suppose the main difference to be investigated here is localized versus regional uplift or so?

2) The introduction is followed directly by the results part. I understand that there are some length issues, but whilst reading it was difficult to follow a red line. Furthermore, figure 1 is cited out of context at the end of the introduction, probably to avoid that the results of figure 2 are the first figure. I suggest to move lines 123-148, somewhat modified, into the introduction or even to a new section related to the study area, linked to figure 1. This would help a lot to introduce the zone of interest and put figure 1 into a more reasonable context.

Minor comments:

Line 28: delete nicely

Line 30 and 32: Be more precise than "this process", as it is not clear which one is meant

Line 33: What would be the alternative mechanism besides transient or long-term?

Line 43: explain what a folding event would be. folding within seconds to minutes? Folding = viscous?

Line 49: inelastic

Line 53: would seismic tremor, aseismic slip, or slow slip be transient or long-term deformation?

Line 64: Maybe have a look at Haghypour et al., 2012, EPSL, where they investigated GPS data versus folded terraces.

Line 71: out of place

Line 160: some English issue here

Line 201: I could not entirely follow why Ainscoe et al. argued that co-seismic slip events are not sufficient based on the sentence before

L231: There are many mechanical models that show synchronous duplex thrusting though. Maybe worth mentioning (e.g. Dal Zilio et al., 2020; Tectonics)

Response to Reviewers

Reviewer #1:

Understanding how the high relief along active mountain fronts (e.g., around the Tibetan Plateau and Tian Shan in Asia) achieved is a key question in continental tectonics. Yet, there exists a long-standing debate on the discrepancy between short-term earthquake uplift and long-term topographic growth, in particular, for fold-and-thrust orogenic belts. In this study, Zhang et al. utilized novel models of developing growth strata and terrace folds by synchronous thrusting in a duplex system. Then the authors applied the model results into the Pishan thrust belt where show obvious discrepancy between positions of long-term topographic highs and transient earthquake uplift as recorded by the high-precision InSAR results. The synchronous thrusting explained very well the discrepant positions of long-term topographic highs (folding crests) and transient maximum earthquake uplift, and provides a new and alternative mechanism for the topographic growth in orogenic belts throughout the world, across different timescales. To my knowledge, this is an exciting and novel finding that will be of broad interest to the research community of continental tectonics, geomorphology, geodesy and even the paleo-environment. Moreover, the manuscript text has been well written, and figures well illustrated. Given the importance and high impact of this study, I strongly support its publication in Nature Communications.

Thank you for your positive evaluation and recommendation of publishing this manuscript in Nature Communications. We have carefully addressed all comments and

suggestions, and thank you again for your thorough revision.

Here I have some minor comments on the figures and the reference format, for the authors to consider.

Fig. 1. The interpretation of thrusting structure under the Keliyang anticline seems to be inconsistent with the offset across the strata. Although this minor problem will not affect the authors' interpretation on the synchronous thrusting under the Pishan belt, it needs to be revised. Please place the horizontal axis labels under the numbers, cite the references for the two earthquakes.

Thanks for pointing this out. We have modified our interpretation of the Keliyang anticline and made this cross-section restorable. The revision, indeed, does not affect our interpretation on the synchronous thrusting under the Pishan belt. We have also done the changes with the figure labels and references suggested by Reviewer 1.

Fig. 2a: "syn-sedimentary model" should be "Growth strata development under synchronous thrusting", Please show horizontal and vertical axis labels in Fig. 2.

We have done the changes suggested by Reviewer 1. Please note that Fig. 2 in the initial manuscript is now Fig. 3 in the revised manuscript.

Fig. 3: The labels for horizontal axes of Fig. 3b & 3c should be placed under the numbers and aligned in the middle.

We have done the changes suggested by Reviewer 1. Please note that Fig. 3 in the

initial manuscript is now Fig. 2 in the revised manuscript.

Fig. 3: Clarify the numbers (3, 4, 5,..., 15, 17, 19 , 21) on the right edge of Fig. 3c in the figure caption, layer numbers, right?

Yes, the numbers on the right edge of this figure denote the layer numbers, and we added a label in the figure for clarification.

Fig. 4: Reorganize the fault legend at the bottom of the figure as: Fault type: active, ceased recently, inactive.

We have changed the fault legend following the suggestion of Reviewer 1.

Also, the format of the references (e.g., Refs. 1 - 3) needs to be double checked and revised according to the journal requirement.

We have carefully checked and revised the format of the references, according to the journal requirement.

Reviewer #2:

Dear editor,

In their paper entitled “Reconciling long-term topographic and transient seismic uplift by synchronous duplex thrusting” Zhang and co-authors investigate how the topography in the southern Tarim basin grew and how it relates to the Pishan

earthquakes (2015, Mw 6.4) and deep duplex structures. The authors mainly conclude that the surface topography (terraces) is due to slip on a deep blind thrust fault of a buried duplex. They also concluded that the recent Quaternary topography was made by the cumulative deformation/uplift of repeated earthquakes on this blind fault.

The question addressed in this study is critical. The deformation model is robust, and model results are convincing. The model nicely explains the growth strata at depth and the geometry of the terraces at surface suggesting that the deformation mechanism proposed by the authors is realistic.

We greatly appreciate your positive comments on the importance of the scientific questions we are addressing here, the robustness of our deformation models and associated results, as well as realistic explanations of the deformation mechanisms.

However, the main conclusion of the paper that: “the topographic growth in Pishan can be produced mainly by repeated individual earthquakes in a duplex system, without the need to invoke additional mechanisms, in certain cases.” is speculative since not supported by any new data/model etc. It moreover contradicts the previous work by Ainscoe et al. (2017). These authors using InSAR, analyses of the same seismic profile together with terraces topography, have already deeply investigated the deformation associated to the Pishan earthquakes and how it relates to the quaternary topography. Yet, they concluded: “over the timescale of multiple earthquake cycles, interseismic and additional aseismic deformation must be invoked”. Yet, to reach this conclusion, Ainscoe et al. modeled the “ground displacements from elastic forward models of earthquakes on the detachment”. Ainscoe et al. were “not able to account for the

anticline by coseismic slip alone”. In contrast, to support their assertion Zhang et al. only speculate (see L212 to 233). Other mechanisms than elastic deformation during repeated earthquakes should be discussed and investigated. This includes interseismic, aseismic deformation, creeping, analestic buckling/folding etc. Those mechanisms are overlooked by Zhang et al. while they could play a major role in explaining the observed uplift of the terraces at surface and the misfit with coseismic displacement observed during the Pishan earthquakes. Consequently, the statement that they “reconcile the discrepancy between the long-term topography of terrace folds and maximum coseismic uplift of the 2015 Mw 6.4 Pishan earthquake on the thrust front” is a bit overrated (so the title of the paper is).

Thank you for these comments. We agree that our description and statements in the original manuscript are not clear enough, which may lead to some misunderstanding for the reviewers. We thank the editors and reviewers for giving us the opportunity to clarify our points, summarized in three aspects below. We have done clarifications in the revised manuscript to reflect our thoughts.

- 1) **We would like to emphasize that we propose a new mechanism, synchronous duplex thrusting, to explain the misalignment between the long-term fold topography and the maximum transient coseismic uplift. This new mechanism will significantly advance our understanding of the topographic growth and mechanics of worldwide fold-and-thrust belts** (e.g., Ainscoe et al., 2017; Elliott et al., 2016; Mackenzie et al., 2016).

We acknowledge that the misalignment between the long-term topography and transient, coseismic uplift has been found previously (Ainscoe et al., 2017; Copley and Reynolds, 2014; Elliott et al., 2016; Mackenzie et al., 2016; Nissen et al., 2007). In particular, Ainscoe et al. (2017) conducted detailed and systematic research on the Pishan belt. As mentioned by the reviewer, they used InSAR datasets to analyze coseismic and postseismic deformation of the 2015 M_w 6.4 Pishan earthquake, and modeled the “*ground displacements from elastic forward models of earthquakes on the detachment*” to obtain the focal parameters of the seismogenic fault. They also analyzed the long-term terrace folds, which are misaligned with the transient deformation (coseismic and postseismic uplift crest). Based on these observations and analyses, Ainscoe et al. (2017) concluded that, they are “not able to account for the anticline by coseismic slip alone”, and “over the timescale of multiple earthquake cycles, interseismic and additional aseismic deformation must be invoked”.

However, Ainscoe et al. (2017) and other previous workers in the Pishan belt mostly only consider slip along a single thrust (frontal ramp fault) for generating the coseismic uplift and long-term terrace folds. Previous research did not realize the possibility of synchronous duplex thrusting, which is observed worldwide fold-and-thrust belts (Boyer, 1992; Butler, 2004; Carrera and Muñoz, 2008; Chen and He, 2022; Mendoza et al., 2019; Tesón and Teixell, 2008), in developing long-term terrace folds and their topographic discrepancy with the maximum coseismic uplift.

This point has been supported by our co-author, Prof. Richard Walker from University of Oxford, who is also a co-author of Ainscoe et al. (2017) and the

director of Ainscoe's PhD thesis, from which the study derived. Thus, Prof. Walker is aware of the content of Ainscoe's paper, and how the discussion was limited to drawing attention to the topographic discrepancy.

We would like to emphasize the key contribution and novelty of our study: **we propose that synchronous duplex thrusting may explain the topographic discrepancy between the long-term folds and maximum coseismic uplift that has been previously found to exist widely within fold-and-thrust belts.** We verify the validity of this new mechanism by developing novel models of terrace fold development and by newly identifying growth strata and modelling their development, which is controlled by deep fault kinematics. These aspects have been highlighted by all three reviewers.

2) Our results and conclusion are supported by new data and new models.

In this study, we have shown the **newly-observed growth strata** based on the interpretation of a seismic reflection profile across the Pishan duplex belt. We also constructed **novel models for development of the growth strata** (Figure R1) **that demonstrated the existence of synchronous thrust slip on the underlying duplex, and then use this to model terrace folding** (Figure R2) **during this process**, using forward modeling with MOVE software (Midland Valley).

These new contributions allow us to propose a new kinematic model for the Pishan duplex belt: two frontal ramp faults have been active synchronously during the in-sequence thrusting. Moreover, the new terrace fold models combined with available

terrace data in this region (Ainscoe et al., 2017; Guilbaud et al., 2017; Xu et al., 2020) further refine our kinematic model of the Pishan belt: the penultimate ramp gradually ceased in the late stage, and all the fault slip was concentrated on the frontal fault ramp. Overall, the newly discovered growth strata and models of growth strata and terrace fold development, combined with published data, support synchronous duplex thrusting in the Pishan duplex system during the in-sequence propagation.

Under the kinematics of synchronous duplex thrusting, in the short term, the seismic slip along frontal ramp fault can generate the maximum uplift above it; in the long term, as new faults develop in the duplex and synchronous thrusting of the penultimate and frontal ramp faults occurs, the locus of topographic uplift will migrate. This will be reflected in the terrace folds above both faults (Figure R3), which shows an obvious topographic discrepancy with the short-term, maximum coseismic uplift, as mentioned above. Therefore, these new data and models support our results and conclusions.

Figure R1. Comparison of growth strata development by fault-related folding under four scenarios: (a) Classic fault-bend fold, (b) Wedge structure, (c) Listric normal fault, and (d) Synchronous thrust faults. The growth strata developed by synchronous thrust faults differ from those by other three models, but resemble the result by the combination of fault-bend fold and the extensional fault. Note: the model

of synchronous thrusting shows similar pre-growth strata geometry with the fault-bend folding and wedge structure thrusting; meanwhile, it shows similar growth strata geometry with listric normal fault.

Figure R2. Models of terrace folding controlled by two ramp faults in a duplex thrust system. (a) Terrace folds are controlled by the frontal ramp fault, as the back ramp was inactive. (b-d) Terrace folds are controlled by synchronous thrusting of two ramp faults in the duplex. Blue-green lines (T0-T4) represent terraces developed at different periods, with T0 for the youngest; dashed red lines denote faults that become gradually inactive, solid red lines show faults that remain active throughout time. Black lines represent inactive faults. See details in the revised Supplemental Materials.

Figure R3. The kinematic relationship in the Pishan fold-and-thrust belt and associated surface deformation. (a) Profiles of river terraces, coseismic and postseismic displacement of the 2015 Pishan earthquake from Ainscoe et al. (2017). The shaded vertical bars show hinge zones of folded terraces. The curved lines represent the modeled terrace deformation above the frontal ramp. (b) Geometry of the Pishan fold-and-thrust belt shows the kinematic configuration with the active and inactive faults, and the Pishan earthquake rupture. The rupture data is from Ainscoe et al. (2017).

3) We do not rule out potential contribution to the ‘topographic discrepancy’ from other factors, and our results do not contradict with Ainscoe et al. (2017)’s work, as discussed in detail below. But we rephrased the title and some statements, for better clarifications (see lines 230-236, 253-261 in the revised text).

Firstly, as we mentioned above, our conclusion highlights that, theoretically, repeated slip (e.g., during earthquake events) along the penultimate and frontal ramp faults, could produce the long-term topography represented by the terrace folds. As

synchronous duplex thrusting has been found to occur within fold-and-thrust belts worldwide (e.g., Boyer, 1992; Carrera and Muñoz, 2008; Chen and He, 2022; Jadoon et al., 2019; Tesón and Teixell, 2008), this process may be the MAJOR mechanism responsible for the ‘topographic discrepancy’ with the transient maximum coseismic uplift.

However, we did not rule out the potential contribution of other factors, e.g., postseismic afterslip and viscoelastic relaxation, aseismic slip, creeping, or pure shear deformation (Copley and Reynolds, 2014; Elliott et al., 2016; Haghypour et al., 2012; Mackenzie et al., 2016; Marinière et al., 2020), for long-term topographic growth. In our study, **the discrepancy between the long-term topographic growth and coseismic uplift reflects the contribution of synchronous duplex thrusting, irrespective of specific slip mechanisms (e.g., seismic versus aseismic) along the faults.** During this process, the terrace folds record all fault slip (coseismic and postseismic) on two fault ramps, while the earthquake uplift represents the result of coseismic fault slip on the frontal fault ramp. Therefore, the crests of coseismic uplift produced from seismic slip along a single ramp fault (in the case of one fault ruptured during earthquake events), will never align with crests of the terrace folds above the duplex thrust system. **Previous studies mainly focused on surface deformation generated by the single fault slip** (e.g., Elliott et al., 2016; Nissen et al., 2007; Taylor, 2016; Whipple et al., 2016), **and did not recognize and analyze the influence of complex kinematic process of underlying faults (e.g., the synchronous thrust faulting proposed in this study) when evaluating the uplift mechanisms for**

generating long-term topography.

Secondly, the mode of slip (e.g., postseismic afterslip, aseismic slip, creeping) is not the determinant factor in resolving the long-term and short-term 'topographic discrepancy', because they will all result of finite displacement along the active faults, and surface deformation above them. The main determinant for the topographic discrepancy is the change in the active fault configuration over time. For instance, it is clear that the frontal ramp fault was responsible for the 2015 Pishan earthquake (Ainscoe et al., 2017; Guilbaud et al., 2017; He et al., 2016; Li et al., 2016; Lu et al., 2016), and the magnitude of topographic growth produced by such slipping is mainly controlled by the geometry of the ramp fault at depth (e.g., Benedetti et al., 2000; Copley and Reynolds, 2014; Ishiyama et al., 2007; McKenzie et al., 2016). However, as we go back in time, the changes in fault array, that is from a single active thrust in the frontal duplex, to the frontal and penultimate faults being active (Figure R3) are the cause of the shift in the location of topographic growth. This would still be the case if the faults were slipping aseisimically, or if a large part of the slip was accrued in the postseismic phase.

Moreover, the contribution of postseismic afterslip to topographic growth may be limited compared to the coseismic uplift. This has been exemplified in the Pishan belt: the postseismic deformation spanning ~1.5 years after the 2015 Pishan earthquake, using InSAR analysis, mainly occurred near the frontal edge of seismogenic, frontal ramp fault (Wang et al., 2020); this observation is consistent with the modeled postseismic deformation in Ainscoe et al. (2017). And the amplitude of the postseismic

deformation only accounts for 1/4 of the maximum coseismic slip (Figure R4) (Wang et al., 2020). The afterslip seismic moment account for even less (~4%) of the coseismic geodetic moment (Wang et al., 2020).

In summary, our research has demonstrated that synchronous duplex thrust slip (through coseismic slip) along two ramp faults alone can produce the long-term terrace folds in the Pishan duplex belt, which are misaligned with the crest of the maximum coseismic uplift. Thus, synchronous duplex thrusting could be the major mechanism responsible for the ‘topographic discrepancy’. Moreover, our study highlights the importance of complexity in the geometry and kinematic process of deep structures in understanding topographic uplift mechanisms.

[REDACTED]

Figure R4 . The distribution of the coseismic and the postseismic slips along the fault, from Ainscoe et al. (2017) and Wang et al. (2020). (a) The coseismic slip distribution; (b) The standard deviation of the slip distribution; (c) The postseismic after-slip distribution at 447 days; (d) The slip distribution of the sum of (a) and (c); The horizontal axis and vertical axis represent fault length along with the strike and fault depth along with the dip, respectively. The length and direction of the arrows indicate the magnitude and direction of slip, respectively. (e) Slip distribution for an afterslip model on a planar, south dipping fault showing contours of coseismic slip in units of meters. (f) Standard deviation of the slip distribution at our preferred smoothing value.

Moreover, the novelty of this study is questionable which may be an obstacle for a publication in Nature Com. Indeed, most of the data presented in Zhang et al.'s paper was actually presented by Anisocoe et al. (2017) and these authors already reached similar conclusions about the link between long term topographic growth, the Pishan earthquakes and deep deformation mechanisms. They said that: (1) "the co-seismic ground displacements from the earthquake do not align with the anticline in Quaternary fluvial terraces", (2) "Our results show that distributed postseismic relaxation of the Cenozoic sediments after the co-seismic stress change also cannot explain the anticline growth" and hence (3) "The remaining mechanisms are therefore slip on different parts of the ramp and flat (particularly interseismic transport on the detachment) and internal deformation of the wedge. Although only a few key faults have been interpreted in Figure 6, both the wedge geometry and the reflections show that it has undergone internal thickening in the past and may still be doing so."

However, it is true that Anisocoe et al. remain elusive regarding the deep mechanisms of deformation within the wedge that are responsible for the Quaternary topographic growth observed at the surface and inferred from terraces topography. Zhang et al explore more deeply these mechanisms using Move software modeling and reached convincing and elegant results. So, the main input of Zhang et al. works is a deformation mechanism at depth.

We would like to take this opportunity to better clarify the novelty of our contribution in this manuscript.

Firstly, we would like to clearly demonstrate that we identified a problem of global significance, i.e., the discrepancy between long-term and short-term topography, based on not just the Pishan, but also from other fold-and-thrust belts. As stated in the introduction of our original manuscript, building the link among long-term surface topography, subsurface deformation, and transient earthquake uplift along worldwide fold-and-thrust belts in active mountain fronts is key to understanding how high-relief topography is achieved (Avouac, 2015; Dal Zilio et al., 2021; Lavé and Avouac, 2001; Melnick, 2016). Currently, studies mainly focus on surface records to unravel the relationship between long-term topographic growth and individual earthquake uplift (Elliott et al., 2016; Nissen et al., 2007). **However, subsurface processes are relatively less understood or generally assumed as the one-to-one correlation between fault displacement and related folds** (Benedetti et al., 2000; Daëron et al., 2007; Lavé and Avouac, 2000; Stockmeyer et al., 2017; Suppe, 1983). **In such an assumption, the discrepancy between long-term topography and coseismic uplift requires other mechanisms for building long-term topography** (Ainscoe et al., 2017; Elliott et al., 2016; Mackenzie et al., 2016). **By integrating new interpretations of growth strata from high-resolution seismic reflection data, novel models of development of this growth strata and models of terrace folding consistent with these results, we propose that that the ‘topographic discrepancy’ can be mainly driven by synchronous duplex thrusting in the active fold-and-thrust belts.**

Secondly, our work does not repeat Ainscoe et al. (2017)’s analysis and

conclusion, as they focused on surface deformation caused by a single earthquake in an attempt to extrapolate this observation back in time. However, the subsurface geometry and kinematics of active structural belts can be complex and have a significant impact on our understanding of surface topographic growth mechanisms. Unfortunately, these aspects remain unexplored in many cases, particularly in regions where seismic reflection profiles are unavailable. Because of this, **our study focused more on mechanisms at depth to explain the ‘topographic discrepancy’.** In their study, Ainscoe et al. (2017) mentioned: (1) *“the coseismic ground displacements from the earthquake do not align with the anticline in Quaternary fluvial terraces”*; (2) *“Our results show that distributed postseismic relaxation of the Cenozoic sediments after the coseismic stress change also cannot explain the anticline growth”*. **We utilize Ainscoe et al. (2017)’s findings as the starting point for our study and use their observations of terrace folding to benchmark our models. From this, we propose the new mechanism, synchronous duplex thrusting, to advance our understanding of the topographic growth process within fold-and-thrust belts.** We have reorganized the text, in particular the Introduction, to correct this misunderstanding.

As mentioned in our manuscript, the duplex thrust system is quite common in the mountain front. Different from traditional one-by-one in-sequence and out-of-sequence thrusting propagation, numerous studies of mountain front duplex thrust systems in North America (Boyer, 1992; Searle et al., 2008; Yin et al., 1989), Andes (Carrera and Muñoz, 2008; Horton, 1999), Pamir (Chen and He, 2022; Thompson et al., 2015), Himalaya (Mendoza et al., 2019), and other regions (Alsop et al., 2018; Butler, 2004;

Tesón and Teixell, 2008), show two or more fault ramps active synchronously (e.g., Figure R5). This synchronous duplex thrusting has also been observed in physical analogue and numerical modeling (Cruz et al., 2010; Dal Zilio et al., 2020; Konstantinovskaya and Malavieille, 2011; Pavlis, 2013; Sun et al., 2016; Wu and McClay, 2011). For example, the numerical models of Dal Zilio et al. (2020) (Figure R6) presented the temporal evolution of a fold-and-thrust belt with two decollements, showing that the fault ramps (e.g., F1, F2, F3) in the duplex system accumulated fault slip throughout the shortening process. This synchronous thrusting in the duplex system is produced to maintain the critical taper angle (Dahlen, 1990; Davis et al., 1983). **Hence, synchronous thrusting is probably widespread in duplex systems around the world highlighting their importance and potential effect on the development of topography.**

Finally, we would like to quote Reviewers 1 and 3, who have provided a succinct statement of the importance of this contribution: Reviewer 1, *“The synchronous thrusting explained very well the discrepant positions of long-term topographic highs (folding crests) and transient maximum earthquake uplift, and provides a new and alternative mechanism for the topographic growth in orogenic belts throughout the world, across different timescales. To my knowledge, this is an exciting and novel finding that will be of broad interest to the research community of continental tectonics, geomorphology, geodesy and even the paleo-environment.”*; Reviewer 3, *“ They demonstrate how a rather simple technique (modelling of growth strata and terraces with Move for example) can help understanding complex problems such as long-term*

versus transient deformation in fold-and-thrust belts. Overall, I think the paper is a nice contribution and may be published, however, after some re-organization of the text."

[REDACTED]

Figure R5. A case study of synchronous thrusting in the Sub-Atlas thrust belt and adjoining basins (Morocco), by Tesón and Teixell (2008). (a-d) The sequential evolution of the Sub-Atlas thrust belt and adjoining basins in selected stages show that two frontal thrusts were active synchronously. See more details in Figure 8 in Tesón and Teixell (2008).

[REDACTED]

Figure R6. Numerical modeling by Dal Zilio et al. (2020) shows the synchronous thrusting during the temporal evolution of the fold-and-thrust belt. The top two panels (modified from Dal Zilio et al. [2020]) show (a) structural evolution and (b) accumulated plastic strain of a fold-and-thrust belt with a basal and an intermediate décollement. The presence of two décollements produces a forward-verging imbrication of shallow ramps at the toe of the wedge, and fault ramps connecting the two décollements propagate from the rear of the model to the outer wedge. (c) Line-drawing of the duplex thrust system in Figure R6a. Fault slips accumulate in the ramps (e.g., F1, F2, and F3) throughout the shortening process, indicating synchronous thrusting of the duplex system.

At last, the organization of the paper is to my opinion awkward. The authors first describe a structural model of duplex and propose different scenarii with different kinematics/timing of fault activation. Syn-tectonics layers are then deposited above the duplex and deformed accordingly to the scenarii. Finally, the authors compare the geometry of these deformed layers to the growth strata observed on the seismic line and terraces geometry measured at the surface. But a logical presentation would have been first to describe the Pishan duplex system and the geometry of the syn-tectonics layers. Then, based on these observations, a structural model the deep duplex structures could be proposed accordingly, and different fault kinematics could be tested. The results of the modeling can be then compared to the reality to discriminate between the scenarii.

Thank you for your comment and suggestion. Reviewer 3 pointed out a similar issue on the text organization. Based on your and Reviewer 3's suggestions, we have done the following revision: first, we introduce the scientific question (the relationship between long-term topographic growth and transient earthquake uplift) and the general background of the 2015 Pishan earthquake. Then, we present the geologic section across the Pishan belt and interpret the seismic reflection profile across the duplex front, newly identifying an interval of growth strata. Based on these observations, we construct a novel model of growth strata development during synchronous thrusting of two adjacent faults in the duplex system. We then take the inferred kinematics of the duplex system and develop novel models of terrace folding which are compared to the results of Ainscoe et al. (2017). Finally, we discussed how the synchronous duplex thrusting reconciles the discrepancy between the long-term topographic growth and the

transient coseismic uplift and its broad implications in mountain growth in active fold-and-thrust belts around the world.

In conclusion, I think the results of this paper about the deep long-term mechanisms of deformation are good and very interesting, even if they are, to my opinion, not well presented/organized. But I don't think this study has reached the level quality requested for publication in Nature Communication because it lacks novelty and, more importantly, robust data/model to support the main conclusion that the topography was made only by repeated earthquakes.

I therefore suggest rejecting the paper.

We would like to express our sincere gratitude for your careful review of our paper and support of our novel models to propose a new kinematic mechanism to generate the transient and long-term topography. We hope that based on the clarifications above and the changes to our manuscript, you can reconsider your recommendation.

Reviewer #3:

Review of the manuscript "Reconciling long-term topographic growth and transient seismic uplift by synchronous duplex thrusting" by Zhang and co-authors, submitted to Nature Communications.

In their paper, the authors establish geometries of growth strata related to ramp activity

in duplex structures and compare them to seismic lines across the Pishan thrust system in the Tarim Basin. Furthermore, they model and investigate terrace folding to track shorter term topographic evolution.

The manuscript is well written and the topic is of interest for a broad scientific community. They demonstrate how a rather simple technique (modelling of growth strata and terraces with Move for example) can help understanding complex problems such as long-term versus transient deformation in fold-and-thrust belts. Overall, I think the paper is a nice contribution and may be published, however, after some re-organization of the text. At times, I had difficulties to follow the argumentation and in my view the study would gain a lot if, for example, the study area (incl. Fig. 1) was introduced earlier.

I hope my comments are useful and constructive

Thank you very much for your overall positive, detailed and very helpful comments and edits, and we have carefully considered them all and address them below.

Comments:

1) The study deciphers differences between long-term and transient deformation. As far as I understood, the authors interpret deformation along duplex thrusts as transient and bulk pure shear deformation of the lithosphere as long-term. Maybe I understood that wrongly, but then it should be explained more clear. Slip along the ramps can also be

aseismic, so I suppose the main difference to be investigated here is localized versus regional uplift or so?

The use of the term *transient* lends itself to confusion and therefore we have changed it in the manuscript. In the study, the coseismic uplift represents the short-term deformation, and thus, the seismically produced topography is short-term topography; in comparison, the terrace folds represent the medium-to-long-term topography and the results of long-term deformation. Both the long-term and short-term deformation is achieved or accumulated by activity of the underlying duplex system. We have clarified them in the revised manuscript (see lines 46-49, 57-59).

About the slip along the ramps: Yes, it may also be aseismic (i.e., postseismic afterslip), but studies of the Pishan earthquake suggest that this fraction of the slip is quite small (<5% of the seismic moment). The key observation is that slip during previous seismic cycles (i.e., on the order of 0.1-1 Myr in the past) can produce the observed terrace folds, which are misaligned with the crest of maximum coseismic uplift from the Pishan earthquake. The main difference investigated here is not the localized versus regional uplift, but rather about the long-term and short-term deformation, and how their topographic signatures are expressed.

2) The introduction is followed directly by the results part. I understand that there are some length issues, but whilst reading it was difficult to follow a red line. Furthermore, figure 1 is cited out of context at the end of the introduction, probably to avoid that the

results of figure 2 are the first figure. I suggest to move lines 123-148, somewhat modified, into the introduction or even to a new section related to the study area, linked to figure 1. This would help a lot to introduce the zone of interest and put figure 1 into a more reasonable context.

Thank you for your comment. Yes, the ‘Results’ part directly follows the ‘Introduction’, which is the journal’s requirement. But we adopted both your and Reviewer 2’s suggestion on the writing structure, and have reorganized the manuscript. First, we introduce the scientific question (the relationship between long-term topographic growth and transient earthquake uplift) and the general background of the 2015 Pishan earthquake. Then, we present the geologic section across the Pishan belt and interpret the seismic reflection profile across the duplex front, newly identifying an interval of growth strata. Based on these observations, we construct a novel model of growth strata development during synchronous thrusting of two adjacent faults in the duplex system. We then take the inferred kinematics of the duplex system and develop novel models of terrace folding which are compared to the results of Ainscoe et al. (2017). Finally, we discussed how the synchronous duplex thrusting reconciles the discrepancy between the long-term topographic growth and the transient coseismic uplift and its broad implications in mountain growth in active fold-and-thrust belts around the world.

Minor comments:

Line 28: delete nicely

We have done the changes.

Line 30 and 32: Be more precise than “this process”, as it is not clear which one is meant

We clarified it with “this synchronous thrusting process”, instead of “this process”.

Line 33: What would be the alternative mechanism besides transient or long-term?

Here we meant to express that our study provides a new kinematic mechanism to reconcile the ‘topographic discrepancy’ between the long-term topographic growth and coseismic uplift; and this mechanism has not been proposed previously for the Pishan and other fold-and-thrust belts. Thus, this mechanism is an alternative one that is different from previous propositions. In more detail, traditionally, mechanisms other than coseismic slip are proposed for long-term topographic growth when the topographic fold is misaligned with the coseismic uplift crust (e.g., Ainscoe et al., 2017; Elliott et al., 2016; Mackenzie et al., 2016). However, our study found that this discrepancy could be produced by synchronous duplex thrusting in the fold-and-thrust belt.

Indeed, our description in the abstract, “*Such a process sheds light on alternative mechanisms of topographic growth in worldwide active fold-and-thrust belts,*” seems a bit confusing. Here, we changed this sentence to “*Thus, our study highlights the importance of characterizing complex subsurface fault kinematics for studying*

topographic growth, and motivates rethinking of the mountain building process in worldwide active fold-and-thrust belts, from short-term to medium-term timescales”.

Line 43: explain what a folding event would be. folding within seconds to minutes?

Folding = viscous?

In this sentence, the folding event equals the earthquake slip within seconds to minutes, for example, the earthquake events in California (Stein and Ekström, 1992; Stein and King, 1984), Montello region in northern Italy (Benedetti et al., 2000), and southern Tianshan foreland (Yao et al., 2020). In order to avoid confusion, we have changed the sentence *“In the short term, topography can be produced via repeated folding and earthquake events.....”* to *“In the short term, topography can be produced via repeated episodes of coseismic folding and uplift, particularly along fold-and-thrust belts, that in aggregate form high-topography mountains.”*

Line 49: inelastic

We have done the changes.

Line 53: would seismic tremor, aseismic slip, or slow slip be transient or long-term deformation?

Yes, these types of deformation could represent the transients of long-term deformation.

Line 64: Maybe have a look at Haghypour et al., 2012, EPSL, where they investigated GPS data versus folded terraces.

Thank you for the suggestion. We have carefully read this paper. This article highlights the contribution of bulk (e.g., aseismic slip) deformation on the regional deformation partitioning and relief of earthquake events based on comparing shortening between the GPS data and folds terraces. It provides valuable material for discussing the mechanism of topographic growth, and we have utilized it in discussing the specific mechanism of fault slip (see lines 293-296).

Line 71: out of place

We have reorganized the manuscript and put the regional geology of the Pishan belt in the third paragraph in the Introduction section.

Line 160: some English issue here

We have reorganized this sentence as *“This proposition is supported by two observations: (1) the sequential younging of growth strata related to anticline formation towards the foreland across the Pishan belt, as documented in this study (Fig. 1d and supplementary Fig. S7) and by Liang et al.³⁷, indicating in-sequence thrusting in this duplex thrust system with recent deformation concentrated on the frontal ramps, and (2) the presence of uplifted terraces above the first two ramps instead of the*

antepenultimate ramp (F4), where the crest of an older fold (the Heshitage anticline) is located (see lines 183-191 in the revised manuscript)”.

Line 201: I could not entirely follow why Ainscoe et al. argued that co-seismic slip events are not sufficient based on the sentence before

By comparing the terrace fold and coseismic uplift crest of the Pishan earthquake, Ainscoe et al. (2017) found that the geometry of folding in the overlying fluvial terraces cannot be fully explained by the repeated coseismic slip, such as the 2015 Pishan earthquake, nor by the early postseismic motion constrained by the InSAR analysis. Therefore, they suggested other mechanisms (e.g., anelastic deformation) are required to build topography in the Pishan fold-and-thrust belt.

Now, we have rewritten this sentence as *“This discrepancy indicated that the terrace folds could not be fully explained by repeated coseismic and postseismic slip in events such as the 2015 Pishan earthquake combined with models correlating surface folds with single seismogenic reverse faults beneath them. This led Ainscoe et al.¹² to suggest that a mechanism other than coseismic slip is required to produce the long-term topography in the Pishan fold-and-thrust belt.”* (see lines 225-230 in the revised manuscript).

L231: There are many mechanical models that show synchronous duplex thrusting though. Maybe worth mentioning (e.g. Dal Zilio et al., 2020; Tectonics)

Thank you for your suggestion. We added more references, including Dal Zilio et al. (2020), to demonstrate that synchronous duplex thrusting is probably widespread in the duplex system around the world. The subsurface kinematic process may be more complex than previously thought, even in a simple duplex thrust system. Furthermore, the widespread-existing synchronous duplex thrusting will motivate the rethinking of topographic growth in high-relief mountains in different time scales.

All Reference used in the Responses

- Ainscoe, E. A., Elliott, J. R., Copley, A., Craig, T. J., Li, T., Parsons, B. E., and Walker, R. T., 2017, Blind Thrusting, Surface Folding, and the Development of Geological Structure in the Mw 6.3 2015 Pishan (China) Earthquake: *Journal of Geophysical Research: Solid Earth*, v. 122, no. 11, p. 9359-9382.
- Alsop, G. I., Weinberger, R., and Marco, S., 2018, Distinguishing thrust sequences in gravity-driven fold and thrust belts: *Journal of Structural Geology*, v. 109, p. 99-119.
- Avouac, J.-P., 2015, Mountain building: From earthquakes to geologic deformation: *Treatise Geophys*, v. 6, p. 381-432.
- Benedetti, L., Tapponnier, P., King, G. C. P., Meyer, B., and Manighetti, I., 2000, Growth folding and active thrusting in the Montello region, Veneto, northern Italy: *Journal of Geophysical Research: Solid Earth*, v. 105, no. B1, p. 739-766.
- Boyer, S. E., 1992, Geometric evidence for synchronous thrusting in the southern Alberta and northwest Montana thrust belts, *Thrust tectonics*, Springer, p. 377-390.
- Butler, R. W. H., 2004, The nature of 'roof thrusts' in the Moine Thrust Belt, NW Scotland: implications for the structural evolution of thrust belts: *Journal of the Geological Society*, v. 161, no. 5, p. 849-859.
- Carrera, N., and Muñoz, J. A., 2008, Thrusting evolution in the southern Cordillera Oriental (northern Argentine Andes): Constraints from growth strata: *Tectonophysics*, v. 459, no. 1, p. 107-122.
- Chen, J., and He, D., 2022, Geometry, kinematics, and mechanism of growth unconformities in the Biertuokuoyi piggyback basin: Implication for episodic growth of the Pamir Frontal Thrust: *Journal of the Geological Society*.
- Copley, A., and Reynolds, K., 2014, Imaging topographic growth by long-lived postseismic afterslip at Sefidabeh, east Iran: *Tectonics*, v. 33, no. 3, p. 330-345.
- Cruz, L., Malinski, J., Wilson, A., Take, W. A., and Hilley, G., 2010, Erosional control of the kinematics and geometry of fold-and-thrust belts imaged in a physical and numerical sandbox: *Journal of Geophysical Research*, v. 115, no. B9.
- Daëron, M., Avouac, J.-P., and Charreau, J., 2007, Modeling the shortening history of a fault tip fold using structural and geomorphic records of deformation: *Journal of Geophysical Research*, v. 112, no. B3.
- Dahlen, F. A., 1990, Critical taper model of fold-and-thrust belts and accretionary wedges: *Annual Review of Earth and Planetary Sciences*, v. 18, no. 1, p. 55-99.
- Dal Zilio, L., Hetényi, G., Hubbard, J., and Bollinger, L., 2021, Building the Himalaya from tectonic to earthquake scales: *Nature Reviews Earth & Environment*, v. 2, no. 4, p. 251-268.
- Dal Zilio, L., Ruh, J., and Avouac, J. P., 2020, Structural Evolution of Orogenic Wedges: Interplay Between Erosion and Weak Décollements: *Tectonics*, v. 39, no. 10.
- Davis, D., Suppe, J., and Dahlen, F. A., 1983, Mechanics of fold-and-thrust belts and accretionary wedges: *Journal of Geophysical Research*, v. 88, no. NB2, p. 1153-1172.

- Elliott, J. R., Jolivet, R., González, P. J., Avouac, J. P., Hollingsworth, J., Searle, M. P., and Stevens, V. L., 2016, Himalayan megathrust geometry and relation to topography revealed by the Gorkha earthquake: *Nature Geoscience*, v. 9, no. 2, p. 174-180.
- Guilbaud, C., Simoes, M., Barrier, L., Laborde, A., Van der Woerd, J., Li, H. B., Tapponnier, P., Coudroy, T., and Murray, A., 2017, Kinematics of Active Deformation Across the Western Kunlun Mountain Range (Xinjiang, China) and Potential Seismic Hazards Within the Southern Tarim Basin: *Journal of Geophysical Research-Solid Earth*, v. 122, no. 12, p. 10398-10426.
- Haghipour, N., Burg, J.-P., Kober, F., Zeilinger, G., Ivy-Ochs, S., Kubik, P. W., and Faridi, M., 2012, Rate of crustal shortening and non-Coulomb behaviour of an active accretionary wedge: The folded fluvial terraces in Makran (SE, Iran): *Earth and Planetary Science Letters*, v. 355-356, p. 187-198.
- He, P., Wang, Q., Ding, K., Wang, M., Qiao, X., Li, J., Wen, Y., Xu, C., Yang, S., and Zou, R., 2016, Source model of the 2015 Mw 6.4 Pishan earthquake constrained by interferometric synthetic aperture radar and GPS: Insight into blind rupture in the western Kunlun Shan: *Geophysical Research Letters*, v. 43, no. 4, p. 1511-1519.
- Horton, B. K., 1999, Erosional control on the geometry and kinematics of thrust belt development in the central Andes: *Tectonics*, v. 18, no. 6, p. 1292-1304.
- Ishiyama, T., Mueller, K., Sato, H., and Togo, M., 2007, Coseismic fault-related fold model, growth structure, and the historic multisegment blind thrust earthquake on the basement-involved Yoro thrust, central Japan: *Journal of Geophysical Research*, v. 112, no. B3.
- Jadoon, S.-U.-R. K., Ding, L., Jadoon, I. A. K., Baral, U., Qasim, M., and Idrees, M., 2019, Interpretation of the Eastern Sulaiman fold-and-thrust belt, Pakistan: A passive roof duplex: *Journal of Structural Geology*, v. 126, p. 231-244.
- Konstantinovskaya, E., and Malavieille, J., 2011, Thrust wedges with décollement levels and syntectonic erosion: A view from analog models: *Tectonophysics*, v. 502, no. 3, p. 336-350.
- Lavé, J., and Avouac, J. P., 2000, Active folding of fluvial terraces across the Siwaliks Hills, Himalayas of central Nepal: *Journal of Geophysical Research Solid Earth*, v. 105, no. B3, p. 5735-5770.
- Li, T., Chen, J., Fang, L. H., Chen, Z. X., Thompson, J. A., and Jia, C. Z., 2016, The 2015Mw 6.4 Pishan Earthquake: Seismic Hazards of an Active Blind Wedge Thrust System at the Western Kunlun Range Front, Northwest Tibetan Plateau: *Seismological Research Letters*, v. 87, no. 3, p. 601-608.
- Lu, R. Q., Xu, X. W., He, D. F., Liu, B., Tan, X. B., and Wang, X. S., 2016, Coseismic and blind fault of the 2015 PishanMw6.5 earthquake: Implications for the sedimentary-tectonic framework of the western Kunlun Mountains, northern Tibetan Plateau: *Tectonics*, v. 35, no. 4, p. 956-964.
- Mackenzie, D., Elliott, J. R., Altunel, E., Walker, R. T., Kurban, Y. C., Schwenninger, J. L., and Parsons, B., 2016, Seismotectonics and rupture process of the MW 7.1 2011 Van reverse-faulting earthquake, eastern Turkey, and implications for

- hazard in regions of distributed shortening: *Geophysical Journal International*, v. 206, no. 1, p. 501-524.
- Melnick, D., 2016, Rise of the central Andean coast by earthquakes straddling the Moho: *Nature Geoscience*, v. 9, no. 5, p. 401-407.
- Mendoza, M. M., Ghosh, A., Karplus, M. S., Klemperer, S. L., Sapkota, S. N., Adhikari, L. B., and Velasco, A., 2019, Duplex in the Main Himalayan Thrust illuminated by aftershocks of the 2015 Mw 7.8 Gorkha earthquake: *Nature Geoscience*, v. 12, no. 12, p. 1018-1022.
- Nissen, E., Ghorashi, M., Jackson, J., Parsons, B., and Talebian, M., 2007, The 2005 Qeshm Island earthquake (Iran)-a link between buried reverse faulting and surface folding in the Zagros Simply Folded Belt?: *Geophysical Journal International*, v. 171, no. 1, p. 326-338.
- Pavlis, T. L., 2013, Kinematic model for out-of-sequence thrusting: Motion of two ramp-flat faults and the production of upper plate duplex systems: *Journal of Structural Geology*, v. 51, p. 132-143.
- Searle, M. P., Law, R. D., Godin, L., Larson, K. P., Streule, M. J., Cottle, J. M., and Jessup, M. J., 2008, Defining the Himalayan Main Central Thrust in Nepal: *Journal of the Geological Society*, v. 165, no. 2, p. 523-534.
- Stein, R. S., and Ekström, G., 1992, Seismicity and geometry of a 110-km-long blind thrust fault 2. Synthesis of the 1982–1985 California Earthquake Sequence: *Journal of Geophysical Research*, v. 97, no. B4, p. 4865.
- Stein, R. S., and King, G. C. P., 1984, Seismic potential revealed by surface folding - 1983 Coalinga, California, earthquake: *Science*, v. 224, no. 4651, p. 869-872.
- Stockmeyer, J. M., Shaw, J. H., Brown, N. D., Rhodes, E. J., Richardson, P. W., Wang, M., Lavin, L. C., and Guan, S., 2017, Active thrust sheet deformation over multiple rupture cycles: A quantitative basis for relating terrace folds to fault slip rates: *Geological Society of America Bulletin*, v. 129, no. 9-10, p. 1337-1356.
- Sun, C., Jia, D., Yin, H., Chen, Z., Li, Z., Shen, L., Wei, D., Li, Y., Yan, B., Wang, M., Fang, S., and Cui, J., 2016, Sandbox modeling of evolving thrust wedges with different preexisting topographic relief: Implications for the Longmen Shan thrust belt, eastern Tibet: *Journal of Geophysical Research: Solid Earth*, v. 121, no. 6, p. 4591-4614.
- Suppe, J., 1983, Geometry and Kinematics of fault-bend folding: *American Journal of Science*, v. 283, no. 7, p. 684-721.
- Taylor, M. H., 2016, Tectonics: Tales of Himalayan topography: *Nature Geoscience*, v. 9, no. 9, p. 649-651.
- Tesón, E., and Teixell, A., 2008, Sequence of thrusting and syntectonic sedimentation in the eastern Sub-Atlas thrust belt (Dadès and Mgoun valleys, Morocco): *International Journal of Earth Sciences*, v. 97, no. 1, p. 103-113.
- Thompson, J. A., Burbank, D. W., Li, T., Chen, J., and Bookhagen, B., 2015, Late Miocene northward propagation of the northeast Pamir thrust system, northwest China: *Tectonics*, v. 34, no. 3, p. 510-534.
- Wang, S., Zhang, Y., Wang, Y., Jiao, J., Ji, Z., and Han, M., 2020, Post-seismic

- deformation mechanism of the July 2015 MW 6.5 Pishan earthquake revealed by Sentinel-1A InSAR observation: *Sci Rep*, v. 10, no. 1, p. 18536.
- Whipple, K. X., Shirzaei, M., Hodges, K. V., and Ramon Arrowsmith, J., 2016, Active shortening within the Himalayan orogenic wedge implied by the 2015 Gorkha earthquake: *Nature Geoscience*, v. 9, no. 9, p. 711-716.
- Wu, J. E., and McClay, K. R., 2011, Two-dimensional analog modeling of fold and thrust belts: Dynamic interactions with syncontractional sedimentation and erosion, K. McClay, J. Shaw, and J. Suppe, eds., *Thrust Fault-Related Folding*.
- Xu, J., Chen, J., Arrowsmith, J. R., Li, T., Zhang, B., Di, N., and Pang, W., 2020, Growth Model and Tectonic Significance of the Guman Fold Along the Western Kunlun Mountain Front (Xinjiang, China) Derived From Terrace Deformation and Seismic Data: *Frontiers in Earth Science*, v. 8.
- Yao, Y., Wen, S., Li, T., and Wang, C., 2020, The 2020 Mw 6.0 Jiashi Earthquake: A Fold Earthquake Event in the Southern Tian Shan, Northwest China: *Seismological Research Letters*, v. 92, no. 2A, p. 859-869.
- Yin, A., Kelty, T. K., and Davis, G. A., 1989, Duplex development and abandonment during evolution of the Lewis thrust system, southern Glacier National Park, Montana: *Geology*, v. 17, no. 9, p. 806-810.

REVIEWERS' COMMENTS

Reviewer #3 (Remarks to the Author):

Secound round of review of the manuscript with the title "Reconciling patterns of long-term topographic growth with coseismic uplift by synchronous duplex thrusting" submitted to Nature Communications.

I checked through the responses and implementations of my previous comments and I am now in favor of accepting the manuscript.

Best wishes

Response to Reviewers

Reviewer #3:

Second round of review of the manuscript with the title "Reconciling patterns of long-term topographic growth with coseismic uplift by synchronous duplex thrusting" submitted to Nature Communications.

I checked through the responses and implementations of my previous comments and I am now in favor of accepting the manuscript.

Best wishes

We really appreciate your time and effort in reviewing our manuscript and are very glad to hear that you are accepting the manuscript. Your feedback has greatly improved the quality of our manuscript.